# Predictive Performance of the FRAX Tool Calibrated for Spain vs. an Age and Sex Model: Prospective Cohort Study with 9082 Women and Men Followed for up to 8 Years

**DOI:** 10.3390/jcm11092409

**Published:** 2022-04-25

**Authors:** Aníbal García-Sempere, Isabel Hurtado, Salvador Peiró, Francisco Sánchez-Sáez, Yared Santaana, Clara Rodríguez-Bernal, Gabriel Sanfélix-Gimeno, José Sanfélix-Genovés

**Affiliations:** 1Foundation for the Promotion of Health and Biomedical Research of Valencia Region (FISABIO), 46020 Valencia, Spain; garcia_ani@gva.es (A.G.-S.); hurtado_isa@gva.es (I.H.); peiro_bor@gva.es (S.P.); sanchez_frasae@gva.es (F.S.-S.); santaana_yar@gva.es (Y.S.); rodriguez_claber@gva.es (C.R.-B.); sanfelix_jos@gva.es (J.S.-G.); 2Spanish Network for Research in Primary Care and Chronicity (RICAPPS), 46020 Valencia, Spain

**Keywords:** FRAX, validation, calibration, epidemiological methods, clinical decision-making, real-world data

## Abstract

In Spain, the Fracture Risk Assessment Tool (FRAX) was adapted using studies with a small number of patients, and there are only a few external validation studies that present limitations. In this prospective cohort study, we compared the performance of FRAX and a simple age and sex model. We used data from the ESOSVAL cohort, a cohort composed of a Mediterranean population of 11,035 women and men aged 50 years and over, followed for up to 8 years, to compare the discrimination, calibration, and reclassification of FRAX calibrated for Spain and a logistic model including only age and sex as variables. We found virtually identical AUC, 83.55% for FRAX (CI 95%: 80.46, 86.63) and 84.10% for the age and sex model (CI 95%: 80.91, 87.29), and there were similar observed-to-predicted ratios. In the reclassification analyses, patients with a hip fracture that were reclassified correctly as high risk by FRAX, compared to the age and sex model, were −2.86%, using either the 3% threshold or the observed incidence, 1.54% (95%CI: −8.44, 2.72 for the 3% threshold; 95%CI: −7.68, 1.97 for the incidence threshold). Remarkably simple and inexpensive tools that are easily transferable into electronic medical record environments may offer a comparable predictive ability to that of FRAX.

## 1. Introduction

By far, hip fracture is the most serious consequence of osteoporosis. It is associated with high financial cost, increased disability, and use of health care services, with patients quickly deteriorating their quality of life and health status [1,2,3]. Every year in Spain, an estimated 45,000 hip fractures occur among people over 65 years of age [4,5]. The incidence of hip fractures drastically increases with age and continues to grow due to global aging. Importantly, effective anti-osteoporotic treatments are available, and the early detection of patients at a higher risk can lead to better prevention and management of fractures and re-fracture [6,7].

Bone mineral density (BMD) is the main criterion used to diagnose osteoporosis, but it is not sufficient to adequately identify patients at a higher risk of suffering an osteoporotic hip fracture. Additionally, many other patient-related factors have been firmly established as risk factors for the occurrence of fragility fractures [8,9,10]. In this context, in the last fifteen years, multiple risk assessment tools based on clinical and personal characteristics have been developed to identify patients at high risk of osteoporotic fracture. Among those is the Fracture Risk Assessment Tool (FRAX), which was released in 2008 and endorsed by several countries and organizations, such as the World Health Organization, the US National Osteoporosis Foundation, or, in our setting, all the relevant Spanish scientific societies (rheumatology, primary care, geriatrics, etc.) and the national and regional authorities. It is the most validated tool in the world, with 26 studies in nine countries [11], and it is probably the most widely used. FRAX is a computer-based algorithm that calculates the 10-year probability of a major osteoporotic fracture (hip, clinical spine, humerus, or wrist fracture) and the 10-year probability of hip fracture. Fracture risk is calculated from 11 variables, including age, body mass index (BMI), and a set of well validated dichotomized risk factors. Femoral neck bone mineral density (BMD) can be optionally input to enhance fracture risk prediction [12].

Since fracture rates are known to vary markedly in different regions of the world [13] and FRAX is based on broad prospective cohort studies, calibration for the FRAX tool should be population-and country-specific. Currently, FRAX is adapted to and available in 65 countries [14]. In Spain, FRAX adaptation was based on results from published information about fracture incidence in three cities and prospective studies from three other cities including a limited number of patients [13], raising doubts about its representativeness. Moreover, notable variations in the prevalence of osteoporosis and fragility hip fractures have been observed in our country, with a contrast between colder, cloudy northern regions and southern and eastern regions with warmer, sunnier Mediterranean weather and different diet patterns [15]. Some studies have evaluated the predictive ability of the FRAX tool with Spanish cohorts, reporting poor performance in two studies but good ability in a third one [16,17,18]. However, these studies have some important limitations, such as the use of limited sample size, short follow-up periods, or being published several years ago; this highlights the need for a robust validation of FRAX in Spain. Finally, there is evidence suggesting that FRAX may predict as well as simpler models that include age and BMD, age and previous fracture, or age, BMD, and vertebral fracture, raising some doubts about the tool’s relative added value [19,20].

Our aim was to externally validate the FRAX tool calibrated for Spain and to compare its performance to that of a logistic model using only age and sex through the data of the ESOSVAL cohort, a cohort composed of a Mediterranean population of 11,035 women and men aged 50 years and over followed for up to 8 years.

## 2. Methods

### 2.1. Setting

The study took place in the region of Valencia (Spain), and the data were obtained from the VHS Integrated Databases (VID). The VID is the result of the linkage, by means of a single personal identification number, of a set of publicly owned, population-based healthcare, clinical and administrative electronic databases in Valencia, which has provided comprehensive information for the region’s five million inhabitants since 2008 [21].

### 2.2. Study Design

In this prospective cohort study, we compared the performance (discrimination, calibration, and reclassification) of two prediction models for hip fracture, the FRAX tool calibrated for Spain and a simple model derived from the ESOSVAL cohort including age and sex computed at the date of the cohort entry (index date) in the ESOSVAL cohort, with a maximum follow-up of 8 years. 

### 2.3. Population

The ESOSVAL study consisted of a large cohort of women and men recruited in the primary health-care centers of the region, with patient follow-up using routine electronic health information systems (VID). It was originally created in 2009 to develop and validate a local population-based prediction scale of osteoporotic fracture applicable to the European Mediterranean population, to be followed for at least 10 years. The ESOSVAL cohort was composed of 11,035 people aged 50 years and over who were recruited by 600 general practitioners and primary care nurses collaborating for free in the ESOSVAL study, attending in 272 primary healthcare centers in the Valencia Health System (VHS) for any health problem between November 2009 and September 2010 and following predefined criteria to attempt to obtain a similar number of men and women with an age distribution as close as possible to the distribution of the region’s population. The ESOSVAL study aims and results obtained, to date, as well as the cohort rationale and characteristics, have been fully described elsewhere [22,23,24,25,26,27,28]. 

### 2.4. Data Sources

The VID includes sociodemographic and administrative data (sex, age, nationality) as well as healthcare information such as diagnoses, procedures, laboratory data, pharmaceutical prescriptions and dispensing, hospitalizations, mortality, healthcare utilization, and public health data. It also includes a set of specific associated databases with population-wide information on significant care areas, such as cancer, rare disease, vaccines, and imaging data [21].

In order to signal the priority of the ESOSVAL study among doctors and to facilitate and enhance the quality of reporting of fracture-related data in VID (for patients included in the ESOSVAL cohort and patients with osteoporosis in general), the Electronic Medical Record (EMR) was modified to include a specific Osteoporosis Risk Sheet (ORS) to register fracture risk factors, monitor patients, and inform diagnostic and therapeutic decision-making [25]. The EMR was modified for all VHS centers. Additionally, doctors and nurses participating in the ESOSVAL project were trained to standardize definitions and to fill in the EMR-specific ORS.

### 2.5. Covariates and Input Variables

We used socio-demographic, clinical, and health services utilization data from VID to describe the cohort and as input variables for the FRAX and the age and sex model. 

### 2.6. Outcomes

The outcome variable was defined as hospitalization for hip fracture (only principal discharge diagnoses), using the International Classification of Diseases 9th revision Clinical Modification [ICD9CM] codes: 820.xx and 733.14 and the ICD10 Spanish adaptation codes: S72.0. S72.1, S72.2 and S72.3. We followed up outcomes from the date of the cohort entry (index dates ranging from 1 November 2009 to 31 September 2010) to 31 December 2017 or to death.

### 2.7. Predictive Tool Risk Computation

Since the current FRAX equations are not published by the authors, we used the FRAX 10-year probability charts calibrated for Spain, stratified by sex, age, body mass index, and clinical risk factors as supplied by the official FRAX site (https://www.sheffield.ac.uk/FRAX/tool.aspx?lang=sp, accessed on 12 January 2022). Data of the patients of the ESOSVAL cohort were introduced by the research team. Only untreated patients were selected. Data on bone mineral density (BMD) measurement were available for only 25.0% of women and 5.2% of men and were not used to estimate FRAX risk.

### 2.8. Analysis

First, we described the baseline characteristics of the ESOSVAL cohort. Second, we used the estimated 10-year risk of hip fracture for each patient using the FRAX tool calibrated for Spain without BMD to obtain the 8-year risk estimates for the ESOSVAL cohort using a logistic regression. We then estimated their risk using a logistic model, including only age and sex as predictors. Individuals were grouped into risk quintiles based on the FRAX and the age and sex model. Third, to assess the calibration of each tool and to compare the average predicted risk with the observed risk during the follow-up period, we estimated the ratio between the fractures expected by each prediction model and those observed over the follow-up period, stratified by quintiles of fracture risk, age, and BMI. We then used calibration slopes representing the observed probability of fracture versus the mean estimated fracture probability for the cohort divided by fifths of estimated probability. In well-calibrated risk prediction systems, predicted and observed rates should track the line of identity [29,30,31,32]. Additionally, the Hosmer-Lemeshow goodness-of-fit test was performed. Fourth, we used receiver-operating characteristic (ROC) curve analyses to assess the predictive performance of the models, with the 95% confidence intervals computed with 2000 stratified bootstrap replicates. In addition, to avoid the overoptimism bias associated with evaluating model prediction performance with the same used to estimate the model, we performed a 10-fold cross-validation [33] (see Appendix A for cross-validation results and the logistic equation of the model). Fifth, we estimated sensitivity, specificity, positive predictive value (PPV), and negative predictive value (NPV) using two different risk threshold values. We used the widely accepted threshold of 3%, as recommended by the Scientific Advisory Council of Osteoporosis in Canada [34] to classify the FRAX scores as 10-years low or high risk of hip fracture in the clinical setting. We also employed the observed incidence, which is model-independent and is commonly used for model assessment purposes as the cut-off point [35]. Sixth, we conducted a reclassification analysis in order to compare the categorization of patients into low-risk and high-risk groups by both tools. To be able to assess the marginal increment of the added variables in FRAX when compared to the age and sex model, we defined FRAX as the “reclassifying” model in the analysis.

## 3. Results

From the 11,035 patients of the ESOSVAL cohort, we included 9082 patients in the study (389 patients were excluded due to lack of BMI information, and 1564 patients were on osteoporotic treatment at the baseline). Of the patients, 59.5% were men, mean age at baseline was 64.2 years old, and 27.3% had secondary or university level studies. A BMI *≥* 30 was observed in 36.2% of patients, 24.3% had diabetes, 8.1% presented ischemic disease, 18.0% were current smokers, and 18.2% experienced at least one fall during the year before the recruitment date (see Table 1).

During the follow-up period, 140 patients with an incident hip fracture were identified over 8 years of follow-up (1.54%, 95 CI%: 1.29, 1.79), of which 57 died before the end of the follow-up (as did 1085 of the 7857 patients without fracture), resulting in a median follow-up of 7.68 years (IQR: 7.60–7.81). The FRAX 10-year estimate of hip fracture in the ESOSVAL cohort was 1.31% (95% CI: 1.25, 1.36), which would result in 119 predicted fractures over 10 years. Table 2 presents the absolute probabilities of hip fracture, which were calculated by each of the models, and the calibration of these probabilities with the absolute fracture rates that were observed over the follow-up period, by quintiles of risk. In FRAX, the model overestimated risk in low-risk patients (0.48 and 0.77 for the first and second risk strata). Observed-to-predicted ratios were higher than 1 for patients at a moderate to high risk (third and fourth quintiles, observed-predicted ratios 1.12 and 1.13, respectively), indicating a slight underestimation of the risk in high-risk patients. For patients at a higher risk, the model showed a marginal overestimation (0.97 for the fifth quintile). The age and sex model underestimated risk in the lowest quintile (1.2) and over-predicted risk in the second and fourth quintiles (0.61 and 0.84, respectively). For patients in the third and fifth quintiles, the model marginally overestimated risk (1.02 and 1.05, respectively). By age strata, FRAX overestimated risk in younger patients (between 65 and 74 years old) and underestimated risk in patients 75 years and older. Calibration by BMI showed that the age and sex model underestimated risk in normal and overweight patients and overestimated risk in obese patients, whereas FRAX showed perfect calibration (see Table 2 and Appendix A). Figure 1 presents a calibration plot for each model, with the observed and predicted rates for each fifth of risk, along with calibration slopes (1.03 for FRAX and 0.94 for the age and sex model). Hosmer-Lemeshow test results were non-significant for both models (2.25, p:0.52 for FRAX; 1.70, p:0.64 for the age and sex model).

By examining their comparative performance, it was seen that both tools had virtually identical AUC, 83.55% for FRAX (CI 95%: 80.46, 86.63) and 84.10% for the age and sex model (CI 95%: 80.91, 87.29), see Figure 2. Using a threshold to define high-risk and indication for the treatment of 3%, FRAX identified 60% (sensitivity) of those who went on to experience a hip fracture, whereas the age and sex model detected 62.86%, as predicted in the index date. The specificity and NPV were high in both tools (specificity: 85.67% and NPV: 99.33% for the age and sex model, 85.51% and 99.27% for FRAX). Using the incidence as a cut-off point value at 1.54%, sensitivity improved to 80% for the age and sex model and 77.14% for the FRAX tool, but specificity decreased (74.54% for FRAX and 75.14% for the age and sex regression, see Table 3).

In the reclassification analysis, the net proportion of patients who experienced a hip fracture and were correctly reclassified as high risk by FRAX, compared to the age and sex model (net reclassification index for events), was −2.86% using both thresholds (95%CI: −8.44, 2.72 for the 3% threshold; 95%CI: −7.68, 1.97 for the incidence threshold). Overall, the change in the proportion of patients assigned a more appropriate risk category for the prediction of hip fracture by FRAX was −3.02%, (95%CI: −8.63, 2.58) using the 3% threshold and −3.46% (95%CI: −8.32, 1.40) using the incidence as a threshold (see Table 4).

## 4. Discussion

Our study included more than 9000 patients of a Mediterranean region followed for an average of almost 8 years, and it directly compared the most used and studied fracture prediction tool to a prediction model using only age and sex. Both models showed high and practically equivalent discriminatory performance as measured by the area under the receiver operating curve. Additionally, in order to identify differences in the discriminative performance for patients with similar risk factors that may not be reflected by AUCs with minimal differences, we further assessed the reclassification of individuals between tools. In examining the value gained from the additional risk factors included in FRAX compared to age and sex, the reclassification analysis showed no advantage in hip fracture prediction over the simpler model. In an analysis of the calibration measures, overall, the models presented similar observed-to-predicted ratios and calibration slopes that were almost identical, showing overall comparable calibration across individuals and among groups of both models. Calibration ratios were also relatively stable in both models.

This study has some strengths. First, we directly compared FRAX, an established fracture prediction tool, to a simple age and sex model in the same population, thus measuring the differences in performance with minimal effect of confounding. Second, we used a relatively large cohort, representative of a south European region of about five million inhabitants, thereby minimizing selection biases. In addition, those who died before the end of the follow-up were included in the analyses, simulating real-world use of the tools. Third, this is, to our knowledge, the most robust validation of the FRAX tool in our setting. Finally, we used age and sex because these variables are universally available and usually very accurate, and we showed that a very simple, easy-to-implement model using only these two variables, offering a comparable predictive ability to that of FRAX, could be transferable to the EMR in VID to timely inform primary care doctors and specialists on the fracture risk of their patients. In an environment of real-world, routinely collected data that can be affected by different types of bias, simpler models using widely available data such as age and sex may have advantages over models requiring many input variables or complex variable definitions. It is important to note, however, that predictive tools are no substitute for clinical judgement, where decisions on treating may be based not only on age and sex considerations but others such as multimorbidity, polypharmacy, or frailty [36].

This study has some limitations. First, we reported results for less than ten years for the follow-up. It has been noted that AUC performance can be potentially affected by follow-up duration. However, this is especially applicable to studies with notably shorter follow-up periods [11,37], whereas our median follow-up was 7.68 years. Second, the VID databases gather real-world clinical practice data and contain information as registered by health professionals during routine clinical practice, but data are not specifically prepared for research. In this sense, studies based on real-world clinical information such as the VID are at risk of well-known biases such a differential recording, misclassification bias, or missing data. However, different interventions (inclusion of an Osteoporosis Risk Sheet in the EMR and a training program to improve risk assessment and registry) ensured an overall high level of completeness of data in the ESOSVAL cohort. Third, as the age and sex model was derived from the ESOSVAL cohort, we cannot rule out the presence of bias in favor of that model when comparing its performance to FRAX in the ESOSVAL cohort. Even if we performed 10-fold cross-validation to correct for overoptimism in the predictive performance of the age and sex model, results concerning the comparative predictive validity of FRAX and the age and sex model would require confirmation in other cohorts. Fourth, although our cohort was representative of the population of the region of Valencia (except for the oldest strata in women, who were slightly underrepresented [25]), it accounted for approximately 11% of the overall Spanish population. Thus, results should be extrapolated with caution, especially given that studies have shown a considerable variability in hip fracture rates and osteoporotic treatment between different regions, even within the same country [27,38].

Thanks to a larger size and longer follow-up, our AUC results for FRAX add robustness to the existing evidence obtained from smaller cohorts in our country [16,17,18]. Internationally, other validation studies of FRAX showed that AUC values for hip fracture prediction ranged from 77% to 83% [39,40,41,42], comparable to the figures that we reported. Many clinical practice guidelines (CPG) addressing the prevention of fragility fracture in patients with osteoporosis are available in Spain. Overall, there is a high variability in the CPG therapeutic recommendations [43], and there is evidence that their uptake in clinical practice is limited [44]. Our study showed that an easy-to-calculate age and sex model, which could be incorporated in the EMR with little effort, has proved to be as helpful and accurate as the FRAX tool to inform doctors about their patient’s risk of osteoporotic hip fracture and guide clinical and prescription decision-making, with no additional burden for them.

In conclusion, we carefully compared the performance of the FRAX tool calibrated for Spain and a logistic model including only age and sex in a large real-world cohort with high-quality data and found no remarkable differences in their predictive ability. In this way, our results call into question the added value in terms of predictive ability of the set of validated risk factors present in FRAX, in addition to age and sex. Real-world, daily clinical practice data derived from electronic clinical databases bring new opportunities to test, assess, and adopt improved diagnostic tools in the clinical setting. If incorporated into EMR, remarkably simple, inexpensive tools, such as the age and sex model we used in our study, have the potential to impact clinical practice way beyond traditional clinical guidelines.

## Figures and Tables

**Figure 1 jcm-11-02409-f001:**
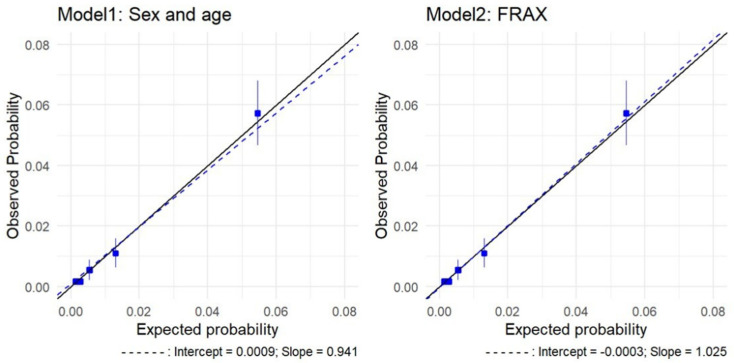
Calibration plots of hip fracture predictions for FRAX and the age and sex model in the comparative analysis: observed risk and average predicted probabilities by probability fifths. Solid line: reference. Blue boxes: quintiles.

**Figure 2 jcm-11-02409-f002:**
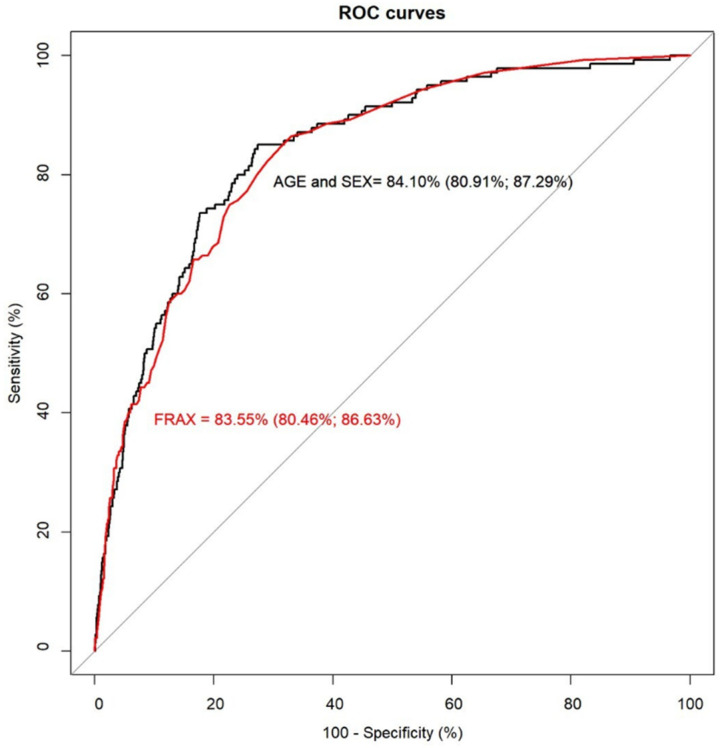
Receiver operating characteristic curves for FRAX and the age and sex model.

**Table 1 jcm-11-02409-t001:** Patient Characteristics at Baseline (*n* = 9082).

	N	%
Gender		
Male	5403	59.5%
Female	3679	40.5%
Age mean (SD)		
	64.2	9.8
Age (strata)		
<65	5274	58.1%
65–69	1329	14.6%
70–74	941	10.4%
75–79	820	9.0%
≥80	718	7.9%
Educational level		
Unknown	641	7.1%
Primary	3632	40.0%
Secondary/University	2481	27.3%
No studies	2328	25.6%
Weight mean (SD)		
Weight	77.6	14
BMI mean (SD)		
BMI	29	4.7
BMI strata		
<20	83	0.9%
20–29	5711	62.9%
≥30	3288	36.2%
Comorbidities		
Osteoporosis	912	10.0%
Diabetes	2210	24.3%
Chronic obstructive pulmonary disease	1007	11.1%
Ischemic disease	735	8.1%
Cerebrovascular disease	379	4.2%
Other chronic conditions	646	7.1%
Smoking		
Current smoker (any quantity)	1633	18.0%
Alcohol Intake		
(>17 U/week for women, >28 U/week for men)	246	2.7%
Menopause before 40 years old		
Yes	235	2.6%
Unknown	5824	64.1%
Calcium intake (mg)		
	806.4	340.2
Calcium intake strata		
Less than 500	1623	17.9%
500–1000	5022	55.3%
1000 or more	2321	25.6%
Unknown	116	1.3%
Sedentary life		
Yes	1501	16.5%
Unknown	157	1.7%
Falls in the previous year (1 or more)		
	1652	18.2%
Untreated hypogonadism		
Yes	247	2.7%
Unknown	697	7.7%
Osteoporosis		
	1840	20.3%
Glucocorticoid use		
(prednisolone equivalent > 5 mg/day at least 3 months in previous year)	79	0.9%
Densitometry		
Densitometric test in ± 24 months from recruitment date	1014	11.2%
Densitometric result		
Normal	337	3.7%
Osteopenia	499	5.5%
Osteoporosis	165	1.8%
Unknown	8081	89.0%
Personal history of osteoporotic fracture		
	538	5.9%
Family history of hip fracture		
	969	10.7%
Calcium and Vit D treatment		
	508	5.6%

Other chronic conditions: renal disease, endocrine diseases, rheumatoid arthritis, chronic liver disease, malabsorption syndrome, prolonged immobility, and organ transplantation.

**Table 2 jcm-11-02409-t002:** Calibration of observed versus predicted hip fracture by FRAX and the age and sex model by quintiles of risk.

	**FRAX**										
	**N**	**Rate**	**Expected Probability**	**[95% Conf. Interval]**		**Observed**	**Expected**	**[95% Conf. Interval]**		**O/E**	**Ratio**
1	1617	0.0006	0.0013	0.0013	0.0013	1	2	2.08	2.11	1/2	0.48 (0.47–0.48)
2	1487	0.0020	0.0026	0.0026	0.0026	3	4	3.89	3.89	3/4	0.77 (0.77–0.77)
3	2014	0.0055	0.0049	0.0048	0.0049	11	10	9.70	9.88	11/10	1.12 (1.11–1.13)
4	2099	0.0143	0.0126	0.0124	0.0128	30	26	26.07	26.81	30/26	1.13 (1.12–1.15)
5	1865	0.0509	0.0524	0.0507	0.0542	95	98	94.55	101.03	95/98	0.97 (0.94–1)
	**AGE and SEX**										
	**N**	**Rate**	**Expected Probability**	**[95% Conf. Interval]**		**Observed**	**Expected**	**[95% Conf. Interval]**		**O/E**	**Ratio**
1	1817	0.0017	0.0014	0.0014	0.0014	3	2	2.46	2.52	**3/2**	1.2 (1.19–1.22)
2	1817	0.0017	0.0027	0.0027	0.0027	3	5	4.85	4.93	**3/5**	0.61 (0.61–0.62)
3	1816	0.0055	0.0054	0.0054	0.0055	10	10	9.75	9.95	**10/10**	1.02 (1.01–1.03)
4	1816	0.0110	0.0131	0.0129	0.0133	20	24	23.43	24.07	**20/24**	0.84 (0.83–0.85)
5	1816	0.0573	0.0545	0.0527	0.0563	104	99	95.71	102.32	**104/99**	1.05 (1.02–1.09)

**Table 3 jcm-11-02409-t003:** Comparison of discriminatory measures using the 3% and the 1.54% thresholds.

	AGE and SEX Model		FRAX	
Threshold	3%	1.54%	3%	1.54%
Specificity	85.67%	75.14%	85.51%	74.54%
Sensitivity	62.86%	80.00%	60.00%	77.14%
Positive predictive value	6.43%	4.80%	6.09%	4.53%
Negative predictive value	99.33%	99.59%	99.27%	99.52%
Correctly classified	85.32%	75.21%	85.11%	74.58%

**Table 4 jcm-11-02409-t004:** Reclassification analysis for FRAX compared to the age and sex model, calculated with the available follow-up.

	NRI [95% CI]	
	3% Threshold	1.54% Threshold
NRI for non-events	−0.17% [−0.67%, 0.34%]	−0.60% [−1.15%, −0.06%]
NRI for events	−2.86% [−8.44%, 2.72%]	−2.86% [−7.68%, 1.97%]
NRI	−3.02% [−8.63%, 2.58%]	−3.46% [−8.32%, 1.40%]

NRI: Net Reclassification Index.

## Data Availability

The datasets presented in this article are not readily available because legal restrictions on sharing the data set apply as regulated by the Valencia regional government by means of legal resolution by the Valencia Health Agency [2009/13312], which forbids the dissemination of data to third parties (accessible at: http://www.san.gva.es/documents/152919/157920/resolucionsolicituddatos.pdf, accessed on 12 January 2022). Upon request, authors can allow access to the databases in order to verify the accuracy of the analysis or the reproducibility of the study. Requests to access the datasets should be directed to the Management Office of the Data Commission in the Valencia Health Agency (email: solicitud_datos@gva.es; telephone numbers: +34-961-928207; +34-961-928198). Requests to access the datasets should be directed to solicitud_datos@gva.es.

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
