# Peer review of "Predictive Performance of the FRAX Tool Calibrated for Spain vs. an Age and Sex Model: Prospective Cohort Study with 9082 Women and Men Followed for up to 8 Years"

_jcm, 2022, doi:10.3390/jcm11092409_

Round 1

Reviewer 1 Report

Please, see below my comments for a minor revision:

Line 39: please change "identifying" into "identify" 

Author Response

Minor concern: Line 39: please change "identifying" into "identify" 

Response: We have modified the manuscript accordingly.

Reviewer 2 Report

The manuscript of García-Sempere et al. ‘Predictive performance of the FRAX tool calibrated for Spain vs. an age and sex model: prospective cohort study with 9,082 women and men followed for up to 8 years.’ is presenting data from a carefully conducted prospective cohort study.

Minor concern:

  • The authors were able to demonstrate, that the value gained from the additional risk factors included in FRAX showed no advantage in hip fracture prediction over the simpler model (age and sex). Unfortunately, the discussion does not highlight this finding. As the introduction clearly states, the FRAX score is a set of well validated dichotomized risk factors, e.g. smoking, steroids, etc., in addition to sex and age.

The authors should try to discuss the impact of the additional risk factors in the context of their findings.

Author Response

Minor concern:

The authors were able to demonstrate, that the value gained from the additional risk factors included in FRAX showed no advantage in hip fracture prediction over the simpler model (age and sex). Unfortunately, the discussion does not highlight this finding. As the introduction clearly states, the FRAX score is a set of well validated dichotomized risk factors, e.g. smoking, steroids, etc., in addition to sex and age.

Response:  we have highlighted this finding in the Discussion section as suggested by the reviewer.

This manuscript is a resubmission of an earlier submission. The following is a list of the peer review reports and author responses from that submission.

Round 1

Reviewer 1 Report

The paper is well written, with a rigorous scientific method.

Suggestions:

The parts highlighted in yellow (lines 32, 34, 56, 66, 73, 138, 190) include suggested corrections (in comments).

Lines 38-40: “Bone mineral density (BMD) is the main criterion used to diagnose osteoporosis, but it has shown poor performance in identifying patients at a higher risk of suffering an osteoporotic hip fracture, having limited sensitivity and specificity.” I would limit a strong statement like this to the fact that BMD alone is not sufficient to diagnose osteoporosis. I would avoid saying that it has poor performance or low sensitivity and specificity.

Line 169 and table 1: There is something that I do not understand. You say that 40.5% were women (line 169). I would suppose that the remaining 59.5% were men…(?).  Why don’t you include men in the table? I could not find, in the material and methods secti, the statement in which you explain that you would show only the data of women.

Discussion: in the discussion section you  should better clarify why you decided to study a simple age and sex model.
